# Circadian Alterations in Brain Metabolism Linked to Cognitive Deficits During Hepatic Ischemia-Reperfusion Injury Using [^1^H-^13^C]-NMR Metabolomics

**DOI:** 10.3390/biomedicines12112536

**Published:** 2024-11-06

**Authors:** Yijing Li, Yanbo Liu, Zhigang He, Zhixiao Li, Hongbing Xiang

**Affiliations:** 1Department of Anesthesiology and Pain Medicine, Hubei Key Laboratory of Geriatric Anesthesia and Perioperative Brain Health, Wuhan Clinical Research Center for Geriatric Anesthesia, Tongji Hospital, Tongji Medical College, Huazhong University of Science and Technology, Wuhan 430030, China; lyj1873600@163.com (Y.L.); bruce_lyb123@hotmail.com (Y.L.); hzgtj2014@tjh.tjmu.edu.cn (Z.H.); 2Key Laboratory of Anesthesiology and Resuscitation, Huazhong University of Science and Technology, Ministry of Education, Wuhan 430030, China

**Keywords:** hepatic ischemia-reperfusion injury, cognitive deficits, circadian alterations, NMR metabolomics, brain metabolism

## Abstract

**Background**: Hepatic ischemia-reperfusion injury (HIRI) is known to affect cognitive functions, with particular concern for its impact on brain metabolic dynamics. Circadian rhythms, as a crucial mechanism for internal time regulation within organisms, significantly influence metabolic processes in the brain. This study aims to explore how HIRI affects hippocampal metabolism and its circadian rhythm differences in mice, and to analyze how these changes are associated with cognitive impairments. **Methods**: A C57BL/6 male mouse model was used, simulating HIRI through hepatic ischemia-reperfusion surgery, with a sham operation conducted for the control group. Cognitive functions were evaluated using open field tests, Y-maze tests, and novel object recognition tests. Magnetic resonance spectroscopic imaging (MRSI) technology, combined with intravenous injection of [2-^13^C]-acetate and [1-^13^C]-glucose, was utilized to analyze metabolic changes in the hippocampus of HIRI mice at different circadian time points (Zeitgeber Time ZT0, 8:00 and ZT12, 20:00). Circadian rhythms regulate behavioral, physiological, and metabolic rhythms through transcriptional feedback loops, with ZT0 at dawn (lights on) and ZT12 at dusk (lights off). **Results**: HIRI mice exhibited significant cognitive impairments in behavioral tests, particularly in spatial memory and learning abilities. MRSI analysis revealed significant circadian rhythm differences in the concentration of metabolites in the hippocampus, with the enrichment concentrations of lactate, alanine, glutamate, and taurine showing different trends at ZT0 compared to ZT12, highlighting the important influence of circadian rhythms on metabolic dysregulation induced by HIRI. **Conclusions**: This study highlights the significant impact of HIRI on brain metabolic dynamics in mice, especially in the hippocampal area, and for the first time reveals the differences in these effects within circadian rhythms. These findings not only emphasize the association between HIRI-induced cognitive impairments and changes in brain metabolism but also point out the crucial role of circadian rhythms in this process, offering new metabolic targets and timing considerations for therapeutic strategies against HIRI-related cognitive disorders.

## 1. Introduction

Clinical liver surgeries, such as interventions for hepatic lesions, trauma surgeries, and liver transplants, often involve the process of ischemia-reperfusion injury (IRI), which can lead to significant liver damage known as hepatic ischemia-reperfusion injury (HIRI). Oxidative stress, calcium overload, mitochondrial dysfunction, activation of liver Kupffer cells and neutrophils, and the release of cytokines and reactive oxygen species (ROS) are all involved in the liver IRI process [1]. These pathophysiological changes can result in liver failure, systemic inflammatory response syndrome (SIRS), and multiple organ dysfunction syndrome (MODS), as well as postoperative cognitive dysfunction [2,3,4,5]. Experimental studies indicate that HIRI primarily affects the central nervous system through mechanisms such as oxidative stress, inflammatory responses, and energy metabolism, contributing to cognitive impairment. Mitochondrial function in hippocampal neurons is damaged by HIRI-induced oxidative stress, which results in energy metabolism disturbances due to the excessive production of ROS and lipid peroxidation products such as malondialdehyde (MDA) [6,7]. ROS are thought to trigger the activation of nod-like receptor protein3 (NLRP3) inflammasome, which in turn causes inflammatory responses and programmed necroptosis of neurons [7,8]. Additionally, HIRI enhances N-methyl-D-aspartate (NMDA) receptor activity via the Src- postsynaptic density protein95 (PSD95)-NR2A signaling pathway, causing over-phosphorylation of the NR2A subunit. This can lead to excitotoxicity and subsequent apoptosis of hippocampal neurons, ultimately resulting in long-term cognitive dysfunction [9]. Circadian rhythms are endogenous biological timing systems that synchronize physiological processes with the day-night cycle. The liver, as a major metabolic organ, is regulated by circadian clock genes, which control its glucose, lipid, and bile acid metabolism, as well as detoxification functions [10,11]. Disruption of circadian rhythms can accelerate the progression of liver diseases, including fatty liver, cirrhosis, hepatitis, and liver cancer [12,13]. Clinical studies have shown that compared to daytime operations, liver transplant surgeries performed at night have a higher incidence of complications and slower recovery times [14]. Attention to circadian rhythms may offer novel perspectives on improving postoperative recovery. Our previous research explored the relationship between HIRI-induced cognitive impairment and gut microbiota dysbiosis, revealing significant reductions in post-HIRI hippocampal acyl-CoA synthetase short-chain family member2 (ACSS2) expression and impaired acetate metabolism related to cognitive decline [15]. A possible relationship between brain metabolic changes and circadian patterns of hippocampal injury was suggested by our observation of day-night fluctuations in cognitive impairment in HIRI-affected C57BL/6 mice, which were connected to changes in lipid metabolism in the hippocampus [16]. Utilizing ^13^C nuclear magnetic resonance (NMR) spectroscopy provides unique advantages in studying brain metabolic processes, allowing for tracking energy substrate utilization and neurotransmitter cycle-related pathways [17,18]. Glucose is the primary energy source for the brain and is mainly metabolized in neurons, while acetate is specifically taken up by astrocytes [17,19]. We infused [2-^13^C]-acetate and [1-^13^C]-glucose to label and detect metabolites in the mouse hippocampus, aiming to investigate metabolic changes and interactions between neurons and astrocytes. Male C57BL/6 mice are not only commonly used in NMR studies but also exhibit significant circadian fluctuations in clock and metabolic genes related to liver metabolism [17,20,21]. These variations are important for processes such as energy and lipid metabolism in the liver. Our research uses male C57BL/6 mice to investigate the alterations in brain energy metabolism and neurotransmitter cycling induced by HIRI in an effort to identify the underlying links to cognitive decline and its diurnal variations.

## 2. Materials and Methods

### 2.1. Animals

Male C57BL/6 mice, aged 6–8 weeks and weighing 20 ± 3 g, were housed at the Experimental Animal Center of Tongji Hospital, Huazhong University of Science and Technology. The housing environment was maintained at a temperature of 23 ± 2 °C and a humidity of 55 ± 10%. Mice were kept on a standard 12-h light/dark cycle (lights on at 8:00 AM, lights off at 8:00 PM) with ad libitum access to food and water. Hepatic ischemia-reperfusion injury surgery was performed at Zeitgeber Time (ZT) 0 (8:00 AM) or ZT12 (8:00 PM). All mice were obtained from Hubei Bent Biological Technology Co., Ltd, Wuhan, China. All experimental protocols were approved by the Institutional Animal Care and Use Committee at Tongji Hospital, Huazhong University of Science and Technology. The experiment was reported according to the ARRIVE guidelines.

### 2.2. Sample Size Estimation and Experimental Grouping and Treatment

Sample size estimation was conducted using the online SPSSAU software (Version 24.0) (https://spssau.com/indexs.html, accessed on 4 November 2024) by selecting the Power analysis for two-factor ANOVA. The calculations were based on an alpha value of 0.05 and a 1-beta value of 0.8. The estimated sample size was 20.673. Considering the 3R principles and the HIRI modeling mortality rate, the initial total sample size for mice was n = 26, leading to a final total sample size of n = 52 for the two infusion substances. Mice infused with [2-^13^C]-acetate were divided into groups as follows: ZT0 Control (CTR) group (n = 6), ZT12 CTR group (n = 6), ZT0 HIRI group (n = 7), and ZT12 HIRI group (n = 7). Similarly, mice infused with [1-^13^C]-glucose were divided into the following groups: ZT0 CTR group (n = 6), ZT12 CTR group (n = 6), ZT0 HIRI group (n = 7), and ZT12 HIRI group (n = 7). The acclimated mice were numbered from lightest to heaviest, and random assignment to the four groups was conducted using a random number table: ZT0 CTR, ZT0 HIRI, ZT12 CTR, and ZT12 HIRI groups. At ZT0 and ZT12, partial hepatic blood flow occlusion was performed on the HIRI group mice to establish the model, while the CTR groups underwent sham operations involving only incision and suture at the same time. All mice subjected to surgery were tested behaviorally on the third day post-operation (72 h of reperfusion) [4,15]. After testing, glucose and acetate were administered, and hippocampal samples were collected for NMR analysis. Except for the technician responsible for HIRI modeling, the researchers conducting behavioral testing, glucose and acetate administration, tissue collection, and NMR data analysis were blinded to group allocations. Post-HIRI modeling, one mouse from the [1-^13^C]-glucose ZT0 HIRI group and two mice from the ZT12 HIRI group died. Additionally, one mouse from the [2-^13^C]-acetate ZT12 HIRI group died. During NMR analysis, due to environmental technical issues, one sample from the [2-^13^C]-acetate ZT12 CTR group and one sample from the ZT0 HIRI group were lost. The experimental flow is illustrated in Figure 1.

### 2.3. The Mouse Model of Hepatic Ischemia–Reperfusion Injury

The HIRI model was established using partial ischemia-reperfusion (involving the left and middle lobes of the liver, with approximately 70% of the total liver area subjected to ischemia) [22,23]. After a 7-day acclimatization period, male C57BL/6 mice underwent surgical procedures. The ZT0 Control and ZT0 HIRI groups were modeled at 8:00 AM, while the ZT12 Control and ZT12 HIRI groups were operated on at 8:00 PM. All mice undergoing modeling were anesthetized preoperatively with 1% sodium pentobarbital administered via intraperitoneal injection (100 mg/kg). Once the mice reached an adequate surgical depth of anesthesia, their limbs were secured to the surgical platform, and the abdominal incision area was thoroughly disinfected. The surgical incision was approximately 2 cm long, initiated 0.5 cm below the xiphoid process, and meticulously opened layer by layer to access the abdominal cavity. Using a retractor to adequately expose the liver, a cotton swab moistened with saline was used to lift the liver, freeing the left lobe, middle lobe, and associated blood vessels while keeping the remaining part of the liver within the abdominal cavity. Micro hemostatic clamps were applied to occlude the respective hepatic artery and vein, and within 5 s, the ischemic liver area changed color from bright red to yellowish-tan, while the liver in the abdominal cavity remained bright red, confirming successful clamping. Sterile saline-soaked gauze was placed over the abdominal incision to ensure tissue hydration, and the mice were placed on a warming blanket to prevent hypothermia. After 90 min, the hemostatic clamps were released, and the ischemic liver rapidly regained its normal color, indicating successful reperfusion. The abdominal cavity was then closed layer by layer with sutures, the wound site was disinfected again, and lidocaine gel was applied to alleviate post-surgical pain. Surgical maneuvers were performed gently to avoid damaging essential abdominal organs and blood vessels. Mice in the CTR group underwent only incision and suture procedures, with postoperative care identical to that of the HIRI group.

### 2.4. Behavioral Test

#### 2.4.1. Open Field Test (OFT)

The Open Field Test (OFT) was conducted following established methods in the literature [24,25] after allowing male C57BL/6 mice to acclimate to their environment. Prior to the experiment, mice were placed in a quiet, dark behavioral laboratory for 2 h to adapt. The open field arena was a gray, opaque polyethylene box (dimensions: 40 × 40 × 40 cm) equipped with a fixed camera to record the mice’s movements throughout the session. The laboratory was maintained in a silent environment with moderate lighting conditions to minimize external factors that could influence the mice’s behavior. At the start of the experiment, each mouse was gently placed in the center of the open field and allowed to move freely for 5 min. The time spent by the mouse in the center of the open field was recorded. At the conclusion of the test, the mouse was returned to its home cage, and medical alcohol was used to clean the equipment to avoid odor interference. Data were recorded and analyzed using ANYMAZE software 7.34 Throughout the experiment, all procedures were performed gently to prevent inducing stress responses in the mice. The OFT was utilized to assess the anxiety levels and locomotor activity of the mice, with anxious mice typically spending less time in the center, which represents the open area of the arena.

#### 2.4.2. Novel Object Recognition Test (NORT)

Prior to the experiment, mice were placed in a quiet, dark behavioral laboratory for a 2-h acclimation period. The experiment was conducted using a gray, opaque polyethylene box, with a camera fixed above to record the mice’s movements throughout the session. The NORT consists of two phases, the learning phase and the testing phase, each lasting five minutes, with a 2-h interval between them. In the learning phase, two identical objects were placed in the open field box for the mice to explore. After this phase, the box was thoroughly cleaned with medical alcohol to eliminate any odors. In the testing phase, one of the identical objects was replaced with a novel object, and the mouse was reintroduced to the box for another 5 min of exploration. Throughout the experiment, ANYMAZE software was used to record and analyze the time spent exploring the novel object (T1) and the familiar object (T2). All procedures were conducted gently to prevent stress responses in the mice. The short-term memory capability of the mice was evaluated using the Recognition Index (RI), calculated as: RI = T2/(T1 + T2). A decrease in RI indicates a decline in the mouse’s short-term memory ability [15,26].

### 2.5. Y-Maze Test

Before the experiment, mice were placed in a quiet, dark behavioral laboratory for a 2-h acclimation period. The Y-maze apparatus consists of three arms (dimensions: 30 × 8 × 15 cm each) arranged at 120° angles to each other. Distinctive patterns are affixed at the end of each arm to help mice memorize spatial locations. The experiment comprises two phases, the learning phase and the testing phase, each lasting five minutes, with a 2-h interval between them. In the learning phase, one arm of the maze (the novel arm) was randomly closed before the experiment. A mouse was placed in one of the other arms (the starting arm) facing the experimenter to prevent pre-exposure to the Y-maze from above. During the learning phase, the mouse was allowed to explore freely between the starting arm and the other open arm for 10 min. Prior to the testing phase, the novel arm was opened, and the mouse was placed back in the Y-maze via the starting arm, allowing it to explore all three arms freely. The apparatus was thoroughly cleaned with medical alcohol between phases to eliminate odor interference. Throughout the entire experiment, data were recorded and analyzed using ANYMAZE software. We recorded the time spent exploring the novel arm to evaluate the mouse’s spatial memory performance [16,27].

### 2.6. Brain Sample Preparation for NMR Study

On the day of the experiment, mice were weighed and then immobilized using a restrainer. A PE-10 catheter with a metal needle tip was inserted into the tail vein of the mouse. Once blood was successfully aspirated into the syringe to confirm patency, the catheter was secured with tape. The syringe end of the PE-10 catheter was connected to a Fusion100 micro-infusion pump, and [1-^13^C]-glucose (enriched at 99%, 0.75 M) was administered intravenously at a volume of 2.2 mL/kg over 2 min. After the infusion was complete, the needle was kept in place for 1 min before being removed, and the mouse was returned to a clean housing cage to move freely for 30 min. Following the same infusion process, [2-^13^C]-acetate (enriched at 99%,1.5 M) was administered at a volume of 3.6 mL/kg [17,28]. Afterward, the mice were deeply anesthetized with 2% isoflurane, followed by swift decapitation. The head was then placed in a microwave oven at 100% power for 14 s. The hippocampus was dissected, weighed, and stored at −80 °C for further analysis [29].

The method for tissue extraction replicated that of our earlier research [30]. In brief, to the frozen sample, HCl/methanol (0.1 mol/L, 100 μL) was introduced and the mixture was homogenized for 1.5 min at 20 Hz using a Tissuelyser II (Qiagen, Hilden, Germany). Subsequently, ice-cold 60% ethanol (800 μL) was added, and the homogenization process was repeated, followed by centrifugation at 14,000× *g* for 10 min. The supernatant was collected. This extraction procedure was performed twice more with 800 μL of 60% ethanol to ensure complete metabolite recovery from the residue. All collected supernatants were then dried using a centrifugal evaporator (Thermo Scientific 2010, Erlangen, Germany) and a freeze vacuum dryer (Thermo Scientific). The resultant dried material was stored for subsequent NMR spectroscopy analysis.

For NMR analysis, the dried sample was reconstituted in 60 μL of D2O (which included an internal standard, 3-(trimethylsilyl) propionic-2, 2, 3, 3-d4 acid sodium salt (TSP, 120 mg/L; 269913-1G, Sigma-Aldrich, Shanghai, China)) and 540 μL of phosphate buffer (pH 7.2). The solution was vigorously vortexed and then centrifuged at 14,000× *g* for 15 min. The clear supernatant was carefully extracted and placed into an NMR tube for analysis.

### 2.7. Acquisition of NMR Spectra

The samples under investigation were analyzed using a Bruker Avance III 600 MHz NMR spectrometer (operating at 298 K) outfitted with an inverse cryoprobe (Bruker BioSpin, Ettlingen, Germany) [31]. Spectra collection was performed utilizing a conventional Watergate pulse sequence. Acquisition parameters for each sample were as follows: p1 (90° pulse length), 8.35 μs; total scans, 256; spectral bandwidth, 20 ppm; pre-scans (dummy scans), 8; data points for free induction decay, 32 K.

### 2.8. NMR Data Processing

All ^1^H-NMR spectra underwent processing and examination using TopSpin software (Version 2.1, Bruker BioSpin) alongside a proprietary software, NMRSpec [32]. Initially, adjustments for phase correction and baseline distortion were manually conducted in TopSpin [31,33]. Subsequently, these adjusted spectra were transferred to NMRSpec for procedures such as aligning spectra, extracting peaks, integrating spectra, and integrating peaks associated with specific chemicals. This software has been applied in numerous metabolomics investigations.

The chemical shift values for key amino acids were found within the 1.20–4.46 ppm range, thus, this interval was selected for detailed analysis. Initially, the peak areas (area under the curve) within this range were automatically quantified for subsequent statistical evaluations. To adjust for varying concentrations, the area of each peak was normalized against the total peak area within this interval in the respective spectrum before proceeding with discriminant analysis [34,35]. The infusion of [1-^13^C]-glucose and [2-^13^C]-acetate in neuronal and astrocytic metabolic pathways is shown in Figure 2 and Figure 3, respectively.

### 2.9. Statistical Analysis

All measurement results are expressed as mean ± standard error of the mean (Mean ± SEM). A significance level of *p* < 0.05 was considered statistically significant. Statistical analyses and graphing were conducted using SPSS17.0 (IBM, USA), R4.2.1, and GraphPad Prism8.0. Behavioral analyses were performed using the Mann–Whitney U test. Comparisons between the ZT0 and ZT12 CTR groups, as well as between the respective CTR and HIRI groups at ZT0 and ZT12, were conducted using a *t*-test. Comparisons of [1-^13^C]-glucose infusion among the four groups (ZT0 CTR, ZT0 HIRI, ZT12 CTR, ZT12 HIRI) were analyzed using two-way ANOVA. Similarly, the comparison of [2-^13^C]-acetate infusion among the four groups was also analyzed using two-way ANOVA. Correlation analyses between metabolic and behavioral data were conducted using Pearson correlation analysis.

## 3. Results

### 3.1. The Day-Night Differences in HIRI-Induced Cognitive Impairment in Mice

In this study, we observed diurnal variations in cognitive impairments in HIRI mice in comparison to the control group of mice (Figure 4). The HIRI mice exhibited compromised spatial and learning memory. In the NORT, the HIRI mice displayed a substantial decline in the novel object RI at both ZT0 and ZT12 compared to CTR mice (Figure 4A, *p* < 0.001, *p* < 0.001, respectively). Additionally, the HIRI mice showed reduced exploration time of the new arm than the CTR mice, regardless of whether they were in the ZT0 or ZT12 group (Figure 4B, *p* < 0.001, *p* < 0.001, respectively). Furthermore, during the OFT, the HIRI mice in both the ZT0 and ZT12 groups spent significantly less time in the center of the open field, indicating anxiety-like behavior (Figure 4C, *p* < 0.001, *p* = 0.001, respectively). Taken together, the behavioral test results suggest impaired learning as well as memory in HIRI mice. Moreover, compared to the ZT0 group of HIRI mice, the spatial learning (*p* < 0.001) and memory abilities (*p* < 0.001) of the ZT12 HIRI group were impaired (Figure 4A,B). In this study, all HIRI mice exhibited a decline in cognitive abilities. Therefore, we utilized ^13^C nuclear magnetic resonance spectroscopy combined with intravenous injection of [2-^13^C]-acetate and [1-^13^C]-glucose in order to analyze the metabolic collaboration between astrocytes and neurons in the brains of HIRI mice.

### 3.2. Infusion with [2-^13^C] Acetate

Mice were administered [2-^13^C] acetate infusions at time points ZT0 and ZT12. Subsequently, the concentration of ^13^C-labeled amino acids in brain tissue extracts was measured using [^1^H-^13^C]-NMR spectroscopy. A t-test was conducted to compare the differences in metabolite levels of astrocytes in mice under physiological conditions at time points ZT0 and ZT12 (Figure 5A, Appendix A), which reveals a notable difference only in the enrichment levels of Glu_4_, which decreased in the ZT12 CTR group compared to the ZT0 CTR group (*p* = 0.046). Further comparison of metabolic changes in hippocampal astrocytes of HIRI mice at ZT0 and ZT12 was made employing the *t*-test. At ZT0, the [2-^13^C]-acetate metabolic results between the CTR group and the HIRI group demonstrated a significant decrease in Glu_4_ (*p* = 0.007) and Gln_4_ (*p* = 0.042) in the HIRI group, along with a notable increase in Myo_5_ (*p* = 0.046, Figure 5B, Appendix A). Other metabolites showed no discernible changes. At ZT12, the HIRI group exhibited a significant decrease in Ala_3_ (*p* = 0.026), whereas levels of Glx_3_ (*p* = 0.021), Myo_5_ (*p* = 0.016), and Myo_3_ (*p* = 0.019) were higher in contrast to the ZT0 CTR group (Figure 5C, Appendix A). Other metabolites showed no significant differences. The effects of time, HIRI, and their interaction on the metabolism of hippocampus astrocytes in mice were examined using two-way ANOVA (Appendix A). There was a marginally significant interaction between time and HIRI on Lac_3_ (F = 4.338, *p* = 0.051, Appendix A), with mean comparisons indicating a decrease in Lac_3_ enrichment in the HIRI group compared to the control at ZT0, but a significant increase at ZT12 post-HIRI (Appendix A). The level of Glx_3_ was impacted by the interaction between time and HIRI (F = 5.873, *p* = 0.026, Appendix A), showing an increase in the HIRI group at ZT0 but a decline compared to the control group at ZT12 (Appendix A).

### 3.3. Infusion with [1-^13^C] Glucose

Our study analyzed the metabolic changes in neurons under physiological conditions in mice at ZT0 and ZT12 using *t*-tests, as illustrated in Figure 6 and Appendix A. At ZT12, compared to the ZT0 CTR group, mice in the CTR group exhibited lower enrichment levels of metabolites such as Ala_3_ (*p* = 0.003), Glu_4_ (*p* = 0.037), Creatine (*p* = 0.039), and Tau_2_ (*p* = 0.020), but higher Gln_4_ (*p* = 0.035) and Glx_2_ (*p* < 0.001) levels (Figure 6A, Appendix A). For HIRI mice, their hippocampal neuron metabolism changes were assessed using T-tests at both ZT0 and ZT12. In the ZT0 CTR versus HIRI group, HIRI mice showed elevated levels of GABA_2_ (*p* = 0.024) and Gln_4_ (*p* = 0.017), with decreased Creatine (*p* = 0.028) enrichment (Figure 6B, Appendix A). The enrichment levels of Ala_3_ (*p* < 0.001), NAA_3_ (*p* = 0.050), GABA_2_ (*p* = 0.030), Glu_4_ (*p* < 0.001), Gln_4_ (*p* = 0.003), GABA_4_ (*p* = 0.002), Creatine (*p* < 0.001), and Tau_2_ (*p* = 0.025) showed significant increases at ZT12 post-HIRI (Figure 6C, Appendix A). A two-factor ANOVA examined the effects of time and HIRI, along with their interaction, on hippocampal neuron metabolism (Figure 6D, Appendix A). Significant interactions were observed for some metabolites, highlighted through mean comparison analyses. The interaction effects on Ala_3_, NAA_3_, Glu_4_, GABA_4_, and Creatine were particularly significant, with distinct changes noted in their levels at ZT0 and ZT12 depending on HIRI influence. Specifically, the interaction between time and HIRI for Ala_3_ showed significance (F = 10.143, *p* = 0.005, Appendix A). At ZT0, there was no significant change in Ala_3_ enrichment, whereas at ZT12, Ala_3_ levels decreased in the HIRI group (Appendix A). The interaction for NAA_3_ was also significant (F = 4.753, *p* = 0.042, Appendix A). At ZT0, there was no significant difference in NAA_3_ enrichment between the CTR and HIRI groups, but at ZT12, NAA_3_ levels were significantly reduced in the HIRI group (Appendix A). For Glu_4_ (F = 17.561, *p* < 0.001, Appendix A), there was no significant difference between the control and HIRI groups at ZT0, but at ZT12, the enrichment level significantly increased under the influence of HIRI (Appendix A). GABA_4_ (F = 7.896, *p* = 0.011, Appendix A) was not affected by HIRI at ZT0, but at ZT12, the levels in the HIRI group were significantly higher than those in the control group (Appendix A). Creatine was influenced by the interaction of time and HIRI (F = 33.169, *p* < 0.001, Appendix A). At ZT0, the enrichment level in the HIRI group was lower compared to the control group, whereas at ZT12, the level increased in the HIRI group (Appendix A).

### 3.4. Correlation Between Hippocampal Metabolites and Cognitive Impairment Behavior in HIRI Mice

Metabolites labeled with [2-^13^C] acetate, including Lac_3_ (r = 0.677), NAA_3_ (r = 0.651), Glx_3_ (r = 0.733), and GABA_2_ (r = 0.643), have been found to positively correlate with the recognitive index in mice. Conversely, Myo_4_ (r = −0.855) shows a negative correlation with RI. Asp_3_ (r = 0.671), Tau_1_ (r = 0.749), and Glx_2_ (r = 0.744) are positively correlated with the exploration time of the novel arm, while Myo_3_ (r = −0.641) is negatively correlated with the exploration time of the novel arm (Figure 7). Similarly, metabolites labeled with [1-^13^C] glucose such as Creatine (r = 0.755) are positively associated with the RI of mice, while Glu_4_ (r = −0.631) and Asp_3_ (r = −0.957) show negative correlations with RI. Additionally, Gln_4_ (r = −0.604), Glx_2_ (r = −0.615), Asp_3_ (r = −0.817), and Myo_5_ (r = −0.837) are negatively correlated with the exploration time of the novel arm. Furthermore, when labeled with [1-^13^C] glucose, GABA_4_ (r = 0.643) is associated with anxiety-related behaviors in mice (Figure 8). On the other hand, Asp_3_ (r = 0.741) and Myo_4_ (r = 0.857) labeled with [2-^13^C] acetate show positive correlations with the time spent in the center of the open field (Figure 7), while Glx_3_ (r = −0.640) and Gln_4_ (r = −0.719) are negatively correlated with the time spent in the center of the open field (Figure 7 and Figure 8).

## 4. Discussion

The metabolic cooperation between neurons and astrocytes plays a crucial role in brain function. An important technique for investigating possible metabolic interactions between astrocytes and neurons linked to cognitive decline is the combination of ^13^C NMR spectroscopy and intravenous infusion of [2-^13^C] acetate, which is primarily metabolized by astrocytes, and [1-^13^C] glucose, which is predominantly consumed by neurons [17,28,36]. Our research findings point out that the HIRI process disrupts the metabolic homeostasis between neurons and astrocytes in the mouse brain, which ultimately causes disturbances in the neuron-astrocyte GABA/glutamate-glutamine cycle and the lactate-alanine cycle. The observed cognitive decline in HIRI mice may be related to brain energy metabolism disorders and changes in neurotransmitter cycling leading to neuroexcitotoxicity. In addition, our study reveals that under physiological conditions in mice, the circadian rhythm has no discernible impact on the interactions between hippocampal neurons and astrocytes. The timing of HIRI surgery, nevertheless, may impact outcomes in mice, with metabolic disruptions in the hippocampus of HIRI mice being more pronounced at ZT12, correlating with the pattern of cognitive impairment—lighter in the morning and more severe at night. Our study identifies possible metabolic pathways and emphasizes the complex metabolic interactions between neurons and astrocytes in maintaining cognitive function.

### 4.1. Cognitive Decline in HIRI Mice: Brain Neurotransmitter Metabolism

In clinical research, alterations in the glutamate-glutamine cycle are regarded as potential biomarkers for several neurodegenerative diseases. In patients with Alzheimer’s disease, metabolic abnormalities in glutamate and glutamine may be associated with cognitive decline [37,38]. Recent evidence from Magnetic Resonance Imaging (MRI) reveals significantly elevated glutamate concentrations in areas such as the basal ganglia and prefrontal cortex of patients with schizophrenia, supporting the hypothesis of excessive glutamatergic neurotransmission [39]. Glutamate, the most prominent excitatory neurotransmitter in the central nervous system, and its precursor, glutamine, are primarily synthesized by astrocytes and transported back to neurons through specific transporters. Under normal brain function, the release and reuptake of glutamate is a dynamic equilibrium. After being released into the synapse by neurons, glutamate is taken up by astrocytes via EAAT1 (glutamate/aspartate transporter, GLAST) and EAAT2 (Glu transporter 1, GLT-1) to prevent excitotoxicity [40]. Subsequently, glutamate is converted into glutamine within astrocytes by glutamine synthetase (GS), which is a crucial process in the glutamate-glutamine cycle [41]. Glutamine is then released extracellularly, allowing neurons to reutilize it, thus completing the cycle [42]. Excessive accumulation of glutamate can impair normal brain function, leading to excitotoxicity and neuronal death. According to studies, high glutamate concentrations activate NMDA receptors, causing a significant influx of calcium ions. Excessive calcium can lead to intracellular calcium overload, triggering apoptosis and necrosis [43]. Under normal circumstances, astrocytes efficiently uptake and metabolize glutamate, maintaining its proper concentration within the synaptic cleft. However, the ability to remove glutamate is reduced when astrocytic function is compromised, leading to increased synaptic concentrations and excitotoxicity [44]. Research suggests that during cerebral ischemia-reperfusion, the dysfunction or downregulation of astrocytic GLT-1 leads to an increased concentration of extracellular glutamate, exacerbating excitotoxicity [45]. In the hippocampal CA1 region of gerbils, GLT-1 and GLAST levels are reduced in the later stages of ischemia-reperfusion events, resulting in extrasynaptic accumulation of glutamate [46]. Our findings indicate that irrespective of time points ZT0 and ZT12, an increase in the concentration enrichment of Gln and GABA in neurons and astrocytes is observed post-HIRI, as depicted in the metabolic diagram. This implies a GABA-Gln cycle between GABAergic neurons and astrocytes that is analogous to the Glu-Gln cycle [47]. The dysfunction of GABAergic neurons disrupts the cortical excitatory/inhibitory balance, leading to core pathological mechanisms of cognitive dysfunction in schizophrenia [48]. Social cognition and spatial learning abilities are hampered when β7 nicotinic acetylcholine receptors in GABAergic neurons are lost [49]. The findings imply that excitatory activation of GABAergic neurons is important in HIRI-induced cognitive impairment. Further variance analysis indicates that neuronal glutamate Glu_4_ enrichment in HIRI mice at time point ZT0 does not significantly differ from the control group. Its level, however, substantially rises at ZT12. On the other hand, no significant effect on Glu_4_ was found in the acetate-labelled astrocyte metabolism at the corresponding ZT12 time point. This demonstrates that nocturnal HIRI surgery may promote Glu cycle disruption between neurons and astrocytes, increase glutamate synthesis, and decrease astrocytic glutamate uptake capacity, which could worsen neuronal excitotoxicity and cause synaptic glutamate accumulation.

Alanine primarily functions in neurotransmitter metabolism as a carrier of ammonia and carbon. It participates in the glutamate-glutamine cycle and plays a role in the lactate-alanine shuttle between astrocytes and neurons. In this shuttle, during the pathway of glutamate synthesis from glutamine in neurons, pyruvate receives NH^4+^ from glutamine via alanine aminotransferase (ALT) to form alanine, which is subsequently taken up by astrocytes. Within astrocytes, alanine is converted back to pyruvate and then reduced to lactate by lactate dehydrogenase (LDH). This lactate is subsequently transported back to neurons, where the ammonia produced in the process contributes to glutamine synthesis in astrocytes, thereby supplementing the brain’s glutamine-glutamate cycle and regulating nitrogen exchange [50]. Our results indicate that the concentration of alanine in hippocampal neurons is affected by time and HIRI. Although no significant change was observed in alanine levels in HIRI mice compared to controls at ZT0 following [1-^13^C]-glucose infusion, there was a notable increase at ZT12. Correspondingly, [2-^13^C] acetate-labelled astrocytes in the hippocampus of HIRI mice showed elevated lactate levels at ZT12. A t-test comparison between the HIRI and control groups at ZT12 indicated reduced alanine enrichment post-HIRI. This illustrates that liver injury at ZT12 may drive neuronal alanine to be transferred to astrocytes and converted to lactate, which then accumulates. However, the unchanged lactate enrichment in neurons at ZT12 despite HIRI suggests a potential disruption in the lactate-alanine cycle between neurons and astrocytes, possibly as a result of the reduced rate of lactate transportation from astrocytes to neurons.

### 4.2. Cognitive Decline in HIRI Mice: Brain Energy Metabolism

Glutamate is essential for maintaining neuronal energy metabolism and cellular functions in addition to the synthesis and metabolism of neurotransmitters. In terms of energy metabolism, glutamate serves as both a neurotransmitter and a carbon source for the tricarboxylic acid (TCA) cycle. It is converted into α-ketoglutarate by glutamate dehydrogenase (GDH) [51,52], a process important for oxidative metabolism in neurons, especially when energy demands are high and where glutamate oxidation provides necessary energy support [53]. Both adult and aged rats have decreased GDH activity during ischemia recovery in their brains, which impacts mitochondrial energy metabolism and raises the possibility that glutamate accumulation is the result of impaired neuronal energy metabolism [54]. Additionally, the regulation of the glutamate-glutamine cycle is influenced by nitrogen oxides and inflammatory factors. Studies have shown that nitric oxide (NO) can regulate glutamate metabolism, affecting the conversion rates of glutamate and glutamine as well as neuronal excitability and energy metabolism [55,56]. Lipopolysaccharide (LPS)-induced neuroinflammation can suppress the expression of EAAT1 and EAAT2 and reduce GS activity, affecting glutamate transport [57]. As previously noted, HIRI at ZT12 dramatically increases glutamate levels in mouse hippocampal neurons, potentially influencing the glutamate-glutamine cycle between neurons and astrocytes through systemic inflammation, leading to excitotoxicity and metabolic dysregulation [58]. Moreover, lactate, which is mostly produced by astrocytes and transported to neurons, serves as an energy substrate and signaling molecule, supporting neuronal metabolism and memory consolidation [59,60]. Disruption of the lactate-alanine shuttle, as previously mentioned, may result in decreased lactate transfer from astrocytes to neurons, causing metabolic disturbances when neurons rely on lactate as an energy substrate.

### 4.3. Cognitive Decline in HIRI Mice: Small Molecule Metabolites

The study highlights significant changes in the concentrations of various low-molecular-weight metabolites, placing a great emphasis on their critical roles in neurological processes. Inositol is an intracellular component whose elevated levels are considered markers of astrocyte proliferation and changes in cell permeability [61]. Astrocytes play a pivotal role in controlling the growth and differentiation of neurons, and they are involved in the clearance of neurotransmitters and the protection of neurons. Clinical research has shown that cognitive impairment in Alzheimer’s Disease patients is positively correlated with levels of myo-inositol in the brain [62]. Our study found significant changes in the content of inositol within astrocytes after HIRI. At ZT12 time points, the inositol content in the HIRI group was higher than that in the control group, indicating an increase in astrocyte proliferation within the brain tissue of mice in the HIRI group. The alteration in the homeostasis of taurine could potentially affect various biological processes, such as osmoregulation, calcium homeostasis, and inhibitory neurotransmitter transmission [63]. Models of chronic neurodegenerative diseases have shown changes in the concentration of taurine in the brain [64,65]. On the other hand, models of insulin-dependent diabetes, insulin resistance, and diet-induced obesity have shown an accumulation of taurine in the hippocampal region [66,67]. Given the potential cytoprotective role of taurine, the accumulation of taurine in the brain may represent a passive activation in response to cognitive impairment events. The increase in neuronal taurine concentrations at the ZT12 time point in HIRI mice suggests that cognitive impairment caused by HIRI may activate the neuroprotective effects of taurine. Creatine is a nitrogenous organic acid that serves as a phosphate energy buffer. Creatine is also one of the main substances regulating osmotic pressure in the brain and is crucial in tissues with high energy demands, such as muscles or the brain. The specific knockout of the creatine transporter gene Slc6a8 results in cognitive deficits [68]. Compared to the control group at ZT0, the concentration of creatine in HIRI mice at the ZT0 time point decreased, while at the ZT12 time point, the neuronal creatine concentration in HIRI mice showed increased, possibly indicating temporal variations in the degree of cognitive impairment caused by HIRI.

## 5. Limitation

Our study utilized only male mice as subjects. However, gender does indeed play a role in the regulation of circadian rhythm genes, with estrogen in females contributing to circadian rhythm regulation [69]. Therefore, further exploration of cognitive impairment induced by HIRI and its circadian differences in female mice would be highly valuable. Our experiments focused solely on the enrichment of metabolites in hippocampal neurons and astrocytes. Considering that the brain is a complex organ with interconnected regions, it is essential for future studies to comprehensively measure more detailed metabolic kinetics across brain regions related to cognitive memory. This approach will aid in understanding the mechanisms of circadian rhythms and HIRI-induced brain metabolic disorders. The study results also indicate changes in anxiety levels in mice. Therefore, the effects of circadian rhythms and HIRI on anxiety-like behaviors in mice warrant further investigation. Measuring hippocampal metabolic changes through a single analytical technique, such as nuclear magnetic resonance, does not sufficiently facilitate mechanistic exploration. It would be beneficial to employ mass spectrometry-based metabolomics, such as liquid chromatography-mass spectrometry (LC-MS), for more detailed pathway analysis.

## 6. Conclusions

For the first time, using high-resolution nuclear magnetic resonance technology and infusions of [1-^13^C] glucose and [2-^13^C] acetate, we explored the changes in hippocampal metabolism and related diurnal variations in HIRI mice. Research findings have unveiled the impact of HIRI on cognitive functions in mice, particularly highlighting the close association between metabolic alterations in the hippocampal region and cognitive deficits. We observed significant changes in brain energy metabolism, neurotransmitter metabolism, and membrane metabolic processes in HIRI mice, indicating a disruption in the metabolic cooperation between astrocytes and neurons. Additionally, the study discovered that HIRI mice exhibited different metabolic and cognitive function changes at various time points (ZT0 and ZT12), revealing a pronounced circadian rhythm in HIRI-induced cognitive impairment. These discoveries offer a new perspective on understanding the mechanisms behind HIRI’s impact on brain metabolism.

## Figures and Tables

**Figure 1 biomedicines-12-02536-f001:**
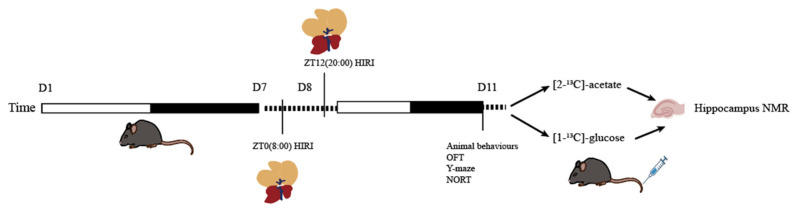
Experimental Flowchart. Abbreviations: NORT, novel object recognition experiment; OFT, open field test; CTR, control; HIRI, hepatic ischemia-reperfusion injury.

**Figure 2 biomedicines-12-02536-f002:**
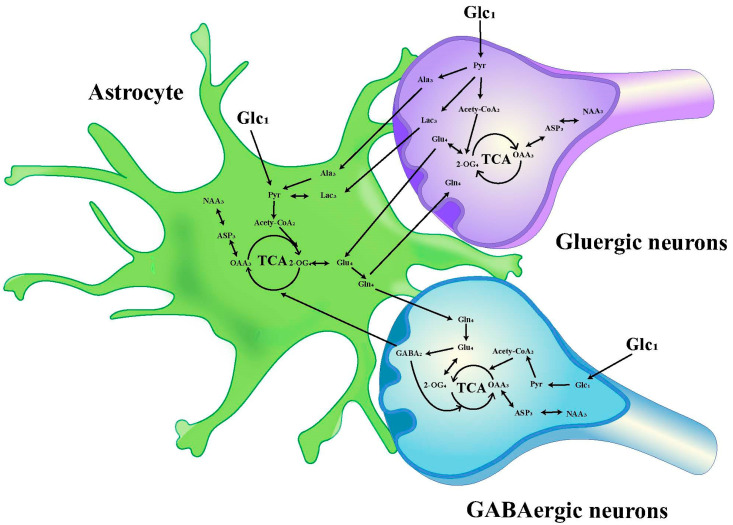
Schematic diagram of ^13^C labeling of metabolites from [1-^13^C] glucose in the first TCA circle between astrocytes, GABAergic neurons and glutamatergic neurons. LactateC3(Lac_3_); AlanineC3(AlaC_3_); γ-aminobutyric acidC2(GABA_2_); GlutamateC4(Glu_4_); GlutamineC4(Gln_4_); Aspartic acidC3(Asp_3_); N-AcetylaspartateC3(NAA_3_).

**Figure 3 biomedicines-12-02536-f003:**
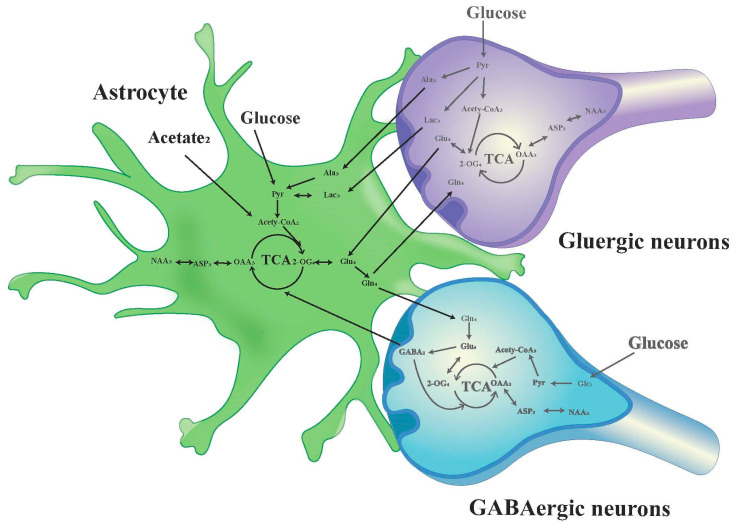
Schematic diagram of ^13^C labeling of metabolites from [2-^13^C] acetate in the first TCA circle between astrocytes, GABAergic neurons and glutamatergic neurons. LactateC3(Lac_3_); AlanineC3(AlaC_3_); γ-aminobutyric acidC2(GABA_2_); GlutamateC4(Glu_4_); GlutamineC4(Gln_4_); Aspartic acidC3(Asp_3_); N-AcetylaspartateC3(NAA_3_).

**Figure 4 biomedicines-12-02536-f004:**
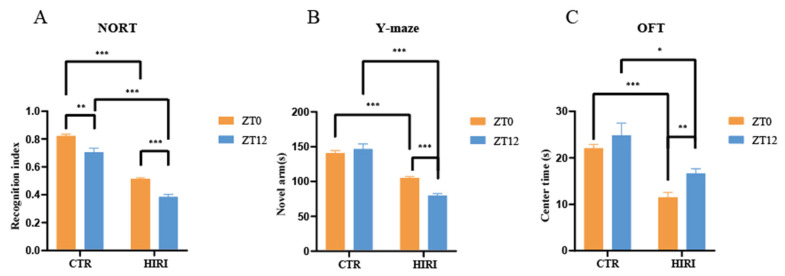
Impact of HIRI on mouse behavior at ZT0 and ZT12. (**A**). The recognition index of NORT. (**B**). The novel arm time of Y-maze test (**C**). The time spent in the center of OFT. Data are presented as the mean ± SEM. * *p* < 0.05, ** *p* < 0.01, *** *p* < 0.001. Abbreviations: NORT, novel object recognition experiment; OFT, open field test; CTR, control; HIRI, hepatic ischemia-reperfusion injury.

**Figure 5 biomedicines-12-02536-f005:**
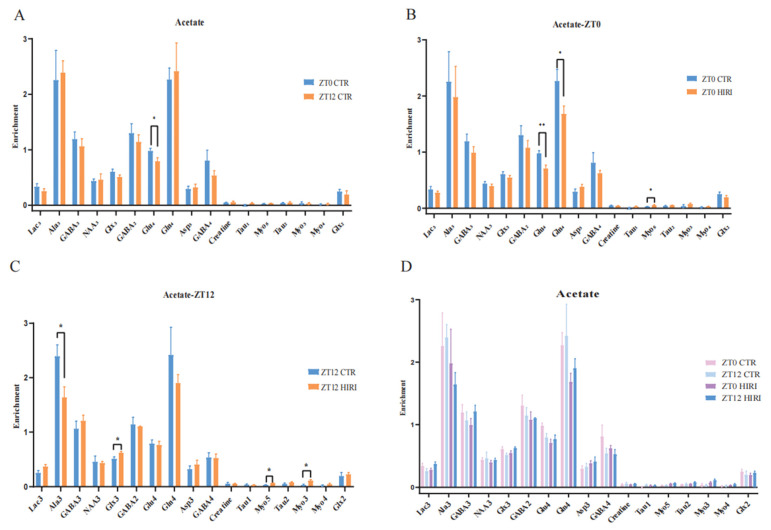
Enrichment of Hippocampal Metabolites with Acetate Infusion (**A**). The enrichment of Lac_3_, Ala_3_, GABA_2_, GABA_4_, GABA_3_, Glu_4_, Gln_4_, Glx_2_, Glx_3_, Asp_3_, Creatine, Ta_u1_, Myo_5_, Tau_2_, Myo_3_, Myo_4_ in ZT0 Ctr group compared to ZT12 CTR group. (**B**). The enrichment of Lac_3_, Ala_3_, GABA_2_, GABA_4_, GABA_3_, Glu_4_, Gln_4_, Glx_2_, Glx_3_, Asp_3_, Creatine, Tau_1_, Myo_5_, Tau_2_, Myo_3_, Myo_4_ in ZT0 Ctr group compared to ZT0 HIRI group. (**C**). The enrichment of Lac_3_, Ala_3_, GABA_2_, GABA_4_, GABA_3_, Glu_4_, Gln_4_, Glx_2_, Glx_3_, Asp_3_, Creatine, Tau_1_, Myo_5_, Tau_2_, Myo_3_, Myo_4_ in ZT12 Ctr group compared to ZT12 HIRI group. (**D**). Comparison of metabolite enrichment in the hippocampal region across four experimental groups. LactateC3(Lac_3_); AlanineC3(AlaC_3_); γ-aminobutyric acidC2(GABA_2_); γ-aminobutyric acidC3(GABA_3_); γ-aminobutyric acidC4(GABA_4_); GlutamateC4(Glu_4_); GlutamineC4(Gln_4_); Glu2+Gln2(Glx_2_); Glu3+Gln3(Glx_3_); Aspartic acidC3(Asp_3_); TaurineC1(Tau_1_); TaurineC2(Tau_2_); Myo-InositolC3(Myo_3_); Myo-InositolC4(Myo_4_); Myo-InositolC5(Myo_5_); CTR, control; HIRI, hepatic ischemia-reperfusion injury. Values represent the mean ± SEM. * *p* < 0.05, ** *p* < 0.01.

**Figure 6 biomedicines-12-02536-f006:**
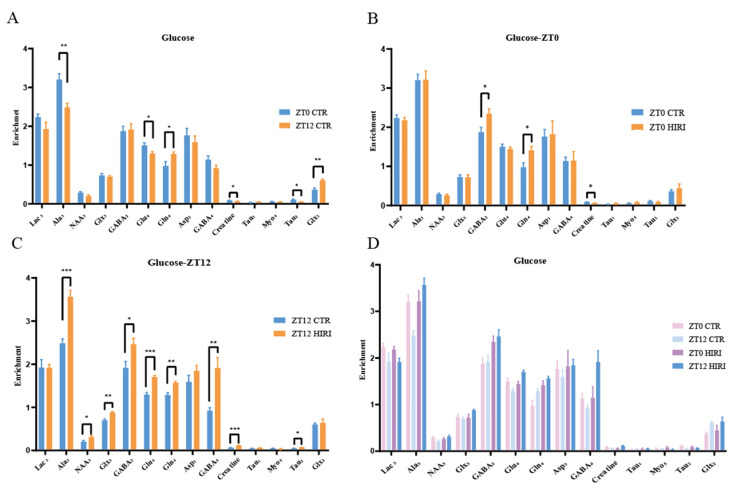
Enrichment of Hippocampal Metabolites with Glucose Infusion (**A**). The enrichment of Lac_3_, Ala_3_, GABA_2_, GABA_4_, Glu_4_, Gln_4_, Glx_2_, Glx_3_, Asp_3_, Creatine, Ta_u1_, Myo_5_, Tau_2_ in ZT0 Ctr group compared to ZT12 CTR group. (**B**). The enrichment of Lac_3_, Ala_3_, GABA_2_, GABA_4_, Glu_4_, Gln_4_, Glx_2_, Glx_3_, Asp_3_, Creatine, Tau_1_, Myo_5_, Tau_2_ in ZT0 Ctr group compared to ZT0 HIRI group. (**C**). The enrichment of Lac_3_, Ala_3_, GABA_2_, GABA_4_, GABA_3_, Glu_4_, Gln_4_, Glx_2_, Glx_3_, Asp_3_, Creatine, Tau_1_, Myo_5_, Tau_2_, Myo_3_, Myo_4_ in ZT12 Ctr group compared to ZT12 HIRI group. (**D**). Comparison of metabolite enrichment in the hippocampal region across four experimental groups. LactateC3(Lac_3_); AlanineC3(AlaC_3_); γ-aminobutyric acidC2(GABA_2_); γ-aminobutyric acidC4(GABA_4_); GlutamateC4(Glu_4_); GlutamineC4(Gln_4_); Glu2+Gln2(Glx_2_); Glu3+Gln3(Glx_3_); Aspartic acidC3(Asp_3_); TaurineC1(Tau_1_); TaurineC2(Tau_2_); Myo-InositolC5(Myo_5_); CTR, control; HIRI, hepatic ischemia-reperfusion injury. Values represent the mean ± SEM. * *p* < 0.05, ** *p* < 0.01, *** *p* < 0.001.

**Figure 7 biomedicines-12-02536-f007:**
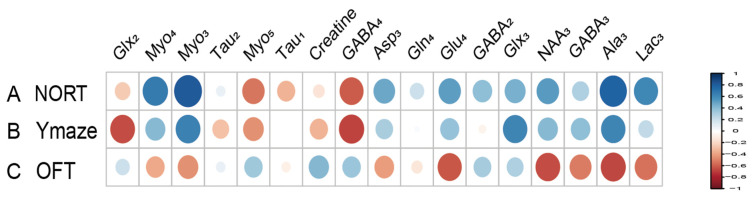
Correlation between [2-^13^C] acetate infusion metabolite concentrations and neurobehavioral cognitive impairment. (A). Correlation between NORT RI and Metabolite Concentrations. (B). Correlation between new arm time in Y Maze and Metabolite Concentrations. (C) Correlation between Center Time in OFT and Metabolite Concentrations. LactateC3(Lac_3_); AlanineC3(AlaC_3_); γ-aminobutyric acidC2(GABA_2_); γ-aminobutyric acidC3(GABA_3_); γ-aminobutyric acidC4(GABA_4_); GlutamateC4(Glu_4_); GlutamineC4(Gln_4_); Glu2+Gln2(Glx_2_); Glu3+Gln3(Glx_3_); Aspartic acidC3(Asp_3_); TaurineC1(Tau_1_); TaurineC2(Tau_2_); Myo-InositolC3(Myo_3_); Myo-InositolC4(Myo_4_); Myo-InositolC5(Myo_5_); NORT, novel object recognition experiment; OFT, open field test; RI, recognition index.

**Figure 8 biomedicines-12-02536-f008:**
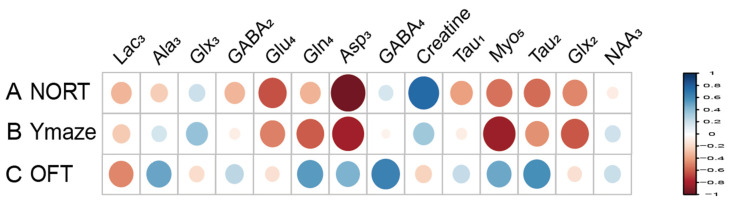
Correlation between [1-^13^C] glucose infusion metabolite concentrations and neurobehavioral cognitive impairment. (A). Correlation between NORT RI and Metabolite Concentrations. (B). Correlation between new arm time in Y Maze and Metabolite Concentrations. (C) Correlation between Center Time in OFT and Metabolite Concentrations. LactateC3(Lac_3_); AlanineC3(AlaC_3_); γ-aminobutyric acidC2(GABA_2_); γ-aminobutyric acidC4(GABA_4_); GlutamateC4(Glu_4_); GlutamineC4(Gln_4_); Glu2+Gln2(Glx_2_); Glu3+Gln3(Glx_3_); Aspartic acidC3(Asp_3_); TaurineC1(Tau_1_); TaurineC2(Tau_2_); Myo-InositolC5(Myo_5_); NORT, novel object recognition experiment; OFT, open field test; RI, recognition index.

## Data Availability

The data that support the findings of this study are available from the corresponding author upon reasonable request.

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
