# Peer review of "Circadian Alterations in Brain Metabolism Linked to Cognitive Deficits During Hepatic Ischemia-Reperfusion Injury Using [1H-13C]-NMR Metabolomics"

_biomedicines, 2024, doi:10.3390/biomedicines12112536_

Round 1
Reviewer 1 Report
Comments and Suggestions for Authors
The authors conducted a study to explore how HIRI affects hippocampal metabolism at different timepoints in the circadian rhythm differences in mice, and to analyze how these changes are associated with cognitive impairment. The authors reported significant associations of HIRI with cognitive impairment and that these differences varied by the timepoints based on circadian rhythm. Suggestions:
1) Please see the ARRIVE reporting guidelines for animal research; and please make sure you've covered each reporting element from the checklist in your manuscript. link: https://www.equator-network.org/reporting-guidelines/improving-bioscience-research-reporting-the-arrive-guidelines-for-reporting-animal-research/
2) In the abstract, please clarify/ briefly explain the meaning of the two timepoints chosen from the circadian rhythm.
3) In the introduction, please state the rationale for choosing the C57BL/6 mice as the animal model in the these experiments; also provide a reference justifying it.
4) Please consider showing a figure which summarizes the methodological steps of your experiment. Were the animals randomized for selection for surgery? Did any animal die? Was there any resultant potential selection bias or any missing data from all the different tests conducted?
5) Lines 91-95: please provide brief description of each test used for behavioral assessments and also provide references for each test (with validity/reliability data for each test)
6) In all figures, please spell out every abbreviation used in the figure in the footnote under the figure. Fig 1 shows two and three asterisks in the figure but the footnote only states the meaning of one asterisk. Please provide complete and comprehensive footnotes for each figure.
7) Please show tables or supplementary tables that provide exact numerical estimates of the data shown in figures or for findings from all major experiments in this manuscript.
8) Please add a limitations paragraph at the end of the discussion section, just before the conclusions para.
Reviewer 2 Report
Comments and Suggestions for Authors
This study used [1H-13C]-NMR metabolomics to investigate the circadian alterations in brain metabolism linked to cognitive deficits during hepatic ischemia-reperfusion injury. This study is interesting with some positive results. I have following concerns.
1. The methods are not in detail. For example, how long did they infused the 13C-labeled acetate or glucose? When did they sacrifice the mice?
2. The statistic methods are in chaos. In figure 1, they used Mann–Whitney U test. However, they did not mention in the method section.
3. The expression of labeled metabolites should be standard. What are Myo3 and Myo5?
4. The abbreviations should be fully spelled.
5. The introduction is not long enough to explain why they used two 13C-labled compounds to investigate the pathways.
6. In the result section, They have to explain why recognition index of control group decreased in ZT0 and ZT12. They also have to explain why the center time of HIRI mice increased.
7. A graph for metabolites pathways of acetate and glucose should be added to explain the metabolite formation.
8. The language should be polished. There are also grammar errors in the figures. For example, “ReEcongnition index” in Figure 1A.
Comments on the Quality of English LanguageNA
Reviewer 3 Report
Comments and Suggestions for Authors
I have reviewed the manuscript titled” Circadian alterations in brain metabolism linked to cognitive two deficits during hepatic ischemia-reperfusion injury using [1H-3 13C]-NMR metabolomics”. This study highlights the impact of Hepatic Ischemia-Reperfusion Injury (HIRI) on brain metabolic in mice, particularly in the hippocampal region, which might be potential metabolic targets for therapeutic interventions against HIRI-related disorders. Overall, the research article is interesting and executed. However, the paper needs a minor revision to enhance its scientific accuracy and clarity. Below are some comments that could help improve the manuscript.
· Why did the author perform the studies on male mice? Due to gender differences, brain metabolic changes in hepatic ischemia-reperfusion injury. Please explain in the manuscript.
· Please include the weight of the mice in the manuscript.
· Why did the authors test the behavior only after 72 hours? If any specific reason, please explain.
· The manuscript lacks the amount of [2-13C] acetate infused in mice and how the dose of [2-13C] acetate was determined. Please include this information in the manuscript.
· The authors did not mention the concentration and volume of [1-13C] glucose infusion in mice. Please include the details in the manuscript with some recent citations.
· In lines 243-244, the authors report, “Our findings indicate that the process of HIRI disrupts the homeostasis of metabolic substances between neurons and astrocytes in mice brains.” Has anyone reported earlier? Please cite to support your results.
· In lines 246-247. the authors noted, " Our study reveals that circadian rhythms have no significant effect on the concentration of brain metabolites under physiological conditions in mice.” Is this finding novel, or has it been reported previously? Please cite earlier reports to support your results.
· In lines 261-262, the authors reported: “Our research indicates that at the ZT12 time point, there is an increase in lactate concentration within astrocytes, a decrease in alanine concentration, and an increase in alanine concentration within neurons”. This sentence is not clear where alanine decreases or increases. Please rephrase it again.
· The discussion section needs improvement. The authors reported several results but did not cite any previous article. The author should compare their finding to prior findings and discuss them in the manuscript.
· Why did the author not perform the metabolomics experiment using mass spectrometry to confirm their finding?
· The author reported that in line 247, circadian rhythms have no significant effect on the concentration of brain metabolites under physiological conditions in mice. What is the possible reason, and have any previous studies reported this type of finding? Please discuss this in the manuscript.
Comments on the Quality of English Languageminor editing
Reviewer 4 Report
Comments and Suggestions for Authors
Comments for improvements
Abstract:
Introduce known mechanisms linking liver injury to brain dysfunction.
Emphasize the implications of the circadian rhythm focus more deeply.
Introduction:
Clarify the connection between circadian rhythms, gut microbiota, and cognitive impairment.
Emphasize the importance of circadian rhythms earlier in the introduction.
Methods:
Provide more details on housing conditions, enrichment, and handling.
Add specifics about the surgical procedure.
Justify the chosen sample size (n=12 per group).
Include more details on behavioral tests (e.g., open-field, Y-maze, novel object preference), covering protocols, parameters measured, and expected outcomes.
Results:
Include P values for all comparisons.
Use consistent terminology for "wild-type mice" or "WT mice."
Explain why time of day (ZT10 vs. ZT12) might influence cognitive abilities.
Provide more insights into why the ZT12 group shows cognitive impairment, possibly due to metabolic or neuroinflammatory factors.
Clarify statistical methods for correlation analysis between metabolites and cognitive impairment.
Explore potential mechanisms behind metabolic changes contributing to cognitive deficits in HIRI mice.
Discussion:
Explain how each specific metabolic disruption (e.g., lactate-alanine shuttle, neurotransmitter imbalance) leads to cognitive decline.
Provide a rationale for why cognitive impairment is more severe at ZT12 than ZT0 in HIRI mice.
Comments on the Quality of English LanguageThe manuscript would benefit from a thorough proofreading to ensure smooth flow and consistency in sentence structure.
Round 2
Reviewer 2 Report
Comments and Suggestions for Authors
No further comments.
Comments on the Quality of English LanguageNA
Reviewer 4 Report
Comments and Suggestions for Authors
Accept in present form.